# Recent Trends in the Epidemiology, Diagnosis, and Treatment of Macrolide-Resistant *Mycoplasma pneumoniae*

**DOI:** 10.3390/jcm11071782

**Published:** 2022-03-24

**Authors:** Tomohiro Oishi, Kazunobu Ouchi

**Affiliations:** 1Department of Clinical Infectious Diseases, Kawasaki Medical School, Kurashiki 701-0192, Japan; 2Department of Medical Welfare for Children, Kawasaki University of Medical Welfare, Kurashiki 701-0193, Japan; kouchi@med.kawasaki-m.ac.jp

**Keywords:** infection, macrolide-resistant *Mycoplasma pneumoniae*, *Mycoplasma pneumoniae*, pediatric

## Abstract

Among *Mycoplasma pneumoniae* (MP) is one of the major pathogens causing lower respiratory tract infection. Macrolide-resistant *Mycoplasma pneumoniae* (MRMP) isolates have been increasing and has become a global concern, especially in East Asian countries. This affects the treatment of MP infection; that is, some patients with MRMP infections fever cannot be controlled despite macrolide therapy. Therefore, alternative therapies, including secondary antimicrobials, including tetracyclines, fluoroquinolones, or systemic corticosteroids, were introduced. However, there are insufficient data on these alternative therapies. Thus, this article provides reviews of the recent trends in the epidemiology, diagnosis, and treatment of MRMP.

## 1. Introduction

*Mycoplasma pneumoniae* (MP) is a major pathogen that causes lower respiratory tract infections, such as pneumonia [1], which might lead to extra-pulmonary diseases [2]. Among MPs, there has been an increase in macrolide-resistant *Mycoplasma pneumoniae* (MRMP), which has become a problem, especially in East Asian countries [3,4,5]. There are varying MRMP rates among countries. MRMP rates have been reported to be higher among children than those among adults [6], and this may be explained by the differences in antibiotic prescriptions in children and adults. Patients with infections caused by MRMP are difficult to treat and tend to have longer durations of infection and higher morbidity. Patients with MRMP might receive delayed appropriate antibiotic treatment and have more severe or prolonged disease courses [7], including extra-pulmonary diseases [8]. Therefore, it is important to investigate the trends in MRMP infections to understand their treatment and prevent their spread. Hence, we reviewed the recent trends in MRMP infections, focusing on their epidemiology, diagnosis, and treatment.

## 2. Mechanisms of Macrolide Resistance for *M. pneumoniae*

*M. pneumoniae* lacks a cell wall. Therefore, beta-lactam antibiotics that target bacterial cell walls are not effective. Macrolides, tetracyclines, and quinolones, which interfere with protein and DNA synthesis, are used to treat MP infections.

Macrolides, which are used especially in children, attach to the 50S ribosomal subunit of the bacterial ribosome. The 50S ribosomal subunit forms the central peptidyl transferase loop in domain V of 23S rRNA [9].

The mechanisms of macrolide resistance and efflux of the antibiotics and target site modification are common, but only the latter has been only found to be associated with macrolide resistance in *M. pneumoniae* [10]. A point mutation in domain V of the 23S rRNA sequence and positions 2063, 2064, and 2067 are the main mutation sites [11]. The A2063G and A2064G transition, which is the most and secondly common, respectively, related with a high level of resistance to 14- and 15-membered macrolides, such as erythromycin (ERY), clarithromycin (CLR), and azithromycin (AZM), but intermediate level of resistance to 16-membered macrolides, such as rokitamycin (RKM) [11]. Mutations of the L22 and L4 ribosomal proteins gene were found, but no macrolide-resistant strains have been isolated [12]. No cross-resistance has been found between macrolide agents and other classes of antibiotics, such as tetracyclines and quinolones. These mutations related to macrolide resistance can be isolated by the PCR as described later.

### Antimicrobial Susceptibility of Macrolide-Resistant M. pneumoniae

Among the point mutations described above, the A2063G transition is the most common, followed by A2064G. The minimum inhibitory concentrations (MICs) of MP isolates with A2063G and A2064G mutations for 14- and 15-membered macrolides were very high (more than 256 g/mL for 14-membered macrolides and 16–64 g/mL for 15-membered macrolides, respectively). However, the MICs of these isolates for 16-membered macrolides are not as high as those for 14- and 15-membered ones, which are 0.0156–16 μg/mL [11]. There are other point mutations in MP isolates, such as the C2617A or C2617T transition, in which the MIC value for macrolides is less than those for A2063G and A2064G mutations (0.0313–8 μg/mL) [10]. The MIC values of lincosamide antibiotics for MPs with A2063G and A2064G mutations were also higher than those for MPs without point mutations. However, these point mutations for macrolide resistance confer no cross-resistance with other antibiotics, such as quinolones and tetracyclines (Table 1) [11,12]. Furthermore, quinolones and tetracyclines resistance have not been seen in MPs even recently [13].

## 3. Epidemiology

The macrolide-resistance rate and the location of 23 rRNA mutations among MP isolates from each country are shown in Table 2 [9,12,13,14,15,16,17,18,19,20]. These data were obtained from reports published after 2015.

The macrolide-resistance rates were different in each region, with the majority of them at 10%, except for Italy and East Asia. The rate in Japan is less than 20% among the East Asian countries. Moreover, the ages are also reported to be a factor of difference in the rate of macrolide resistance. Specifically, the rate of MRMP among children tend to be higher than that that among adults [6]. Most of the point mutations in domain V of 23S rRNA were A2063G mutations as previously described.

Recently, the number of cases with MP infections had decreased remarkably due to the pandemic of COVID-19, and the decreasing rate of MRMP has also been reported [21].

## 4. Diagnosis

If we use macrolide agents for the patients with MP infections, they defervesce generally within 48–72 h after initiation of macrolide treatment [28]. However, the fevers possibly continue more than 48–72 after macrolide treatment among the patients with MRMP infections. That is, the continuous fever might give an opportunity to suspect MRMP infections.

As definite diagnosis, bacterial cultures are the standard for many kinds of bacterial infections. However, this procedure is difficult to perform due to its complicated and time-consuming nature for MP infections. In particular, MP requires special agar or broth and takes more than two weeks to grow [29]. Therefore, many methods for detecting MP antigens have been used in clinical laboratories for diagnosis, such as polymerase chain reaction (PCR) and real-time PCR (rt-PCR) using respiratory samples, such as nasopharyngeal and specimens. The target genes for PCR and rt-PCR differ; 16S RNA and P1 gene are used for PCR and rtPCR, respectively. These tests are highly sensitive and specific, and there are kits available for detecting not only the MP antigen but also point mutation related to macrolide resistance [30] (Table 3). However, special techniques and equipment are required to perform PCR; thus, the tests are not easily accessible. Recently, kits for immunochromatography, including rapid antigen tests for the detection of MP, have been developed [31,32,33]. Antigen tests can provide rapid results and can be easily performed by anyone, but their sensitivities are not as high as PCR. Thus, it was not possible to detect point mutations related to macrolide resistance in MP. Recently, some point-of-care assays have been used. It is a combination of PCR and a quenching probe (Q Probe), in which the target DNA hybridized with a fluorescence-labeled oligonucleotide is detected by fluorescence quenching, and the turn-around time is approximately two hours. They can be used as point-of-care testing devices since they can detect not only MP genes but also MR mutations, such as A2063G and A2064G mutations [34,35]. These devices have been approved for use with health insurance in Japan.

## 5. Treatment

### 5.1. Antibiotics

There are four clinical practice guidelines for antibiotic therapy for MP infection [1,36,37,38]. In addition to the guidelines of the Pediatric Infectious Diseases Society (PIDS) and the Infectious Diseases Society of America (IDSA) for the management of community-acquired pneumonia in infants and children older than 3 months, published in 2011 [1], antimicrobial therapies for MRMP have been described. [29,30,31]. Two Japanese guidelines were devised: (1) guidelines for the Management of Respiratory Infectious Diseases in Children by the Japanese Society for Pediatric Infectious Diseases, the Japanese Society of Pediatric Pulmonology in 2011 and 2017 [36], and the (2) Guiding Principles for Treating *Mycoplasma pneumoniae* pneumonia (MPP) of the Committee of the Japanese Society of Mycoplasmology in 2014 [37]. The guidelines state that macrolides are recommended as the first-line drug of choice for the treatment of MPP, and the efficacy of macrolides may be assessed with relatively high accuracy in the presence or absence of defervescence within 48–72 h after initiation of macrolide treatment as already described. In addition, the use of tosufloxacin or tetracyclines may be considered when necessary for patients with pneumonia who do not respond to macrolides. However, tetracyclines are contraindicated in children younger than 8 years of age.

Lung et al. published recommendations for the management of community-acquired pneumonia (CAP) in children in 2016, stating that (1) physicians should consider MRMP if children with *M. pneumoniae*-associated CAP fails to respond to macrolide therapy; (2) doxycycline is recommended for the treatment of MRMP-associated CAP in children aged > 8 years; (3) for children ≤ 8 years old infected with MRMP-associated CAP, doxycycline should be used when the benefit outweighs the risk; and (4) fluoroquinolone is an alternative option to doxycycline for MRMP-associated CAP in children ≤ 8 years old.

The three guidelines recommend tetracycline and quinolone agents as alternative antibiotics when treating patients with MRMP infections. These antimicrobial agents are described not only in the above three guidelines but also in the guidelines from PIDS and IDSA in the U.S. Specifically, they recommend doxycycline (for patients aged > 7 years) or levofloxacin/moxifloxacin (for adolescent patients who have reached skeletal maturity) as second-line oral drugs and levofloxacin as a second-line parenteral drug for MPP [1].

There are few reports on the effectiveness of tetracyclines for MRMP infections [28,39,40,41]. Tetracyclines are more effective than macrolides in the treatment of MRMP infections. However, they have a side effect of tooth enamel hypoplasia and discoloration of permanent teeth [41], and thus, they should not be prescribed to patients less than 7 or 8 years of age before permanent teeth have erupted.

Fluoroquinolones (FQs) are also alternative antibiotics for the treatment of MRMP infections. However, they have been reported to damage the cartilage of the weight-bearing joints of juvenile animals [42]. It has been reported that Achilles tendon injuries associated with FQs have an incidence of 0.08–2.0% [43]. As a result of FDA regulations, fluoroquinolones can only be prescribed to patients with complicated infections and for whom there is no suitable alternative antibiotic [44]. Therefore, few reports specify that fluoroquinolones are more effective than macrolides for defervescence after treatment of MRMP infections [28,45,46].

Tosufloxacin (TFLX) has been approved for the treatment of children with MP infection in Japan since 2010. There are two reports on the evaluation of TFLX in children with MP infections [43,44]. It is effective clinically and also has a good rate of eradication of MP. The eradication rate of MP was 100% (6/6) for children with MP infections, including two isolated MRMPs treated with TFLX, whereas it was 42.8% (3/7) for those isolated MPs, including two isolated MRMPs (0/2) who were treated with clarithromycin [47]. There have been no reports on permanent arthropathy caused by TFLX in Japanese children since its approval in Japan.

Infection with MRMP can result in extra-pulmonary diseases, as already mentioned [8]. In these cases, the pharmacokinetic (PK) parameters, such as lipid solubility, are very important. AZM has a very high lipid solubility among antibiotics used for MP. Blood–brain barrier (BBB) permeability is also an important PK parameter for central nervous system (CNS) infections, such as encephalitis. MN, LVX, and MXF have a high BBB permeability; therefore, they are the therapeutic agents used for CNS infections due to MRMP [48]. 

### 5.2. Therapy Other Than Antibiotics: Systemic Corticosteroids

MRMP infections are difficult to treat and tend to cause prolonged fever due to the host immune responses [48,49]. Therefore, if patients with MP infections do not defervesce regardless of antibiotic therapy, corticosteroid therapy can be selected as an alternative therapy to reduce host immune responses. Higher levels of inflammatory cytokines are observed in patients with MRMP than in those with macrolide-susceptible *Mycoplasma pneumoniae* [50].

However, there are some controversies about not only the necessity but also the object and time of initiation or the dose of corticosteroid therapy. The Japanese guidelines state, “systemic administration of corticosteroids may be considered for patients with serious pneumonia, although it should be reserved for patients who do not respond to appropriate antimicrobial treatment” [36]. Specifically, systemic administration of corticosteroids should be considered in cases of severe pneumonia with fever lasting for ≥7 days and lactate dehydrogenase of >480 IU/L [51]. Regarding the time of initiation, You et al. reported that intravenous methylprednisolone therapy was effective in patients who showed persistent fever for 36–48 h or disease progression [52]. In a meta-analysis of clinical efficacy and safety of high (10–30 mg/kg) and low (1–2 mg/kg) doses of methylprednisolone in the treatment of children with severe MP pneumonia, it was reported that high-dose methylprednisolone was effective for these patients without increasing the incidence of adverse effects [53]. MP infections due to MRMP tended to be severe because of the difficulty in treatment; therefore, it is important to establish a method of systemic corticosteroid therapy for these patients.

## 6. Conclusions

The rates of MRMP are different across countries; MRMP infections were highly prevalent in East Asian by the mid-2010s, and difficulty in achieving defervescence in patients with MRMP infections has been a major challenge. Therefore, some guidelines have been published for the treatment of MP infection, including MRMP infections. Specifically, alternative antibiotics or systemic corticosteroids should be considered if patients with MP infections do not defervesce despite macrolide therapy. Although no concrete method of systemic corticosteroids treatment has been established yet, and no alternative antibiotic has yet been developed, a rise in resistant MP may have occurred due to inappropriate antibiotic use. Moreover, due to the pandemic of COVID-19, the rate of MRMP seems to have decreased in accordance with the decreasing number of cases of MP infections. Whereas antibiotics are possibly used to prevent pneumonia with bacterial superimposition, macrolide might be also used as nonantimicrobial actions [54]. However, the method of diagnosis has improved, enabling the immediate diagnosis and prescription of appropriate antibiotics for MRMP. The establishment of diagnostic and therapeutic criteria for MRMP infections is important, as are efforts to decrease MRMP incidence.

## Figures and Tables

**Table 1 jcm-11-01782-t001:** MICs of macrolide and other antibiotics for *M. pneumoniae* isolated from patients and reference strains; (µg/mL) erythromycin (ERY), clarithromycin (CLR), azithromycin (AZM), rokitamycin (RKM), spiramycin (SPM), lincomycin (LCM), and clindamycin (CLI). ND: not done. Adapted from Morozumi M. et al. (2010) [11] and Cao B. et al. (2010) [12].

Mutation in 23 S rRNA Gene	ERY	CLR	AZM	RKM	LVX	MXF	TET	MN
A2063G *1 (*n* = 96) *2 (*n* = 41)	32 to >64	32 to > 64	16 to > 64	0.0156–16	0.5–1	0.0625–0.125	ND	0.0625–1
	128 to >256	64 to 1256	2–32	ND	0.125–2	0.008 to 0.032	0.032–0.5	0.016–0.5
A2064G *1 (*n* = 7) *2 (*n* = 4)	64 to >64	16 to >64	16–64	8–16	0.5–1	0.0625–0.125	ND	0.0313–1
	256	256	4–8	ND	0.5–0.25	0.032	0.125–0.25	0.064–0.125
A2063T *2 (*n* = 1)	32	16	0.064	ND	0.25	0.032	0.25	0.25
A2063C *1 (*n* = 1)	>256	>256	16	4	ND	ND	ND	ND
C2617A *1 (*n* = 1)	1	0.5	0.0313	0.0313	1	0.125	ND	1
C2617T *1 (*n* = 1)	8	1	0.0313	0.0625	ND	ND	ND	ND

*1: Date from [11]. The strains from Japan in 2002–2008. *2: Date from [12]. The strains from China in 2008–2009.

**Table 2 jcm-11-01782-t002:** Macrolide resistance rate in *M. pneumoniae* clinical isolates.

Country	Year	% of Macrolide Resistance (Number of Resistant Strains or *M. pneumoniae*-Positive Specimens/Total Strains or Specimens Tested)	23S rRNA Mutations (%)	Reference
USA				
USA (Ohio)	2015–2019	2.8% (14/485)	A2063G (78.6%) A2064G (21.4%)	Lanatai et al., 2021 [15]
USA (9 states)	2012–2018	8.3% (37/446)	A2063G (86.5%)	Xiao et al., 2020 [16]
ASIA				
Taiwan (Kaohsiung)	2016–2019	54.3% (44/81)	A2063G (100%)	Chang et al., 2021 [3]
Japan (2 institutions)	2015–2016	56.3% (85/151)	A2063G (97.7%) A2064G (2.3%)	Kawakami et al., 2021 [4]
Japan (Ibaraki)	2016–2017	65.3% (174/226)	ND	Akashi et al., 2018 [5]
Taiwan (2 institutions)	2017–2019	77% (90/100)	A2063G (82.8%) A2063T (16.1%) A2064G (1.1%)	Hung et al., 2021 [17]
China (Shanghai)	2016–2019	56.1% (60/107)	A2063G (100%)	Zhou et al., 2020 [9]
Korea (4 institutions)	2019–2020	78.5% (73/93)	A2063G (98.6%) A2063T (1.4%)	Lee et al., 2021 [18]
Japan	2019–2020	11.3% (6/53)	A2063G (100%)	Morozumi et al., 2020 [19]
Japan	2018	14.3% (4/29)	A2063G (100%)	Nakamura et al., 2021 [20]
China	2020–2021	7.7% (6/78)	A2063G (100%)	Chen J. et al. 2022 [21]
EUROPE				
Spain (Barcelona)	2013–2017	8.0% (11/137)	A2063G (63.6%) A2064G (18.2%) A2064T (9.1%) C2617A (9.1%)	Rivaya et al., 2020 [22]
Italy (12 institutions)	2013–2015	20.0% (3/15)	A2063G (66.7%) A2064G (33.3%)	Loconsole et al., 2019 [23]
Sweden (4 institutes)	2016–2017	1.6% (1/61)	A2063G (100%)	Gullsby et al., 2019 [24]
Finland (Kmenlaakso)	2017–2018	0% (0/11)	-	Kurkela et al., 2019 [25]
England (London)	2010–2015	9.3% (4/43)	A2063G (100%)	Brown et al., 2015 [26]
Slovenia	2006–2016	0.8% (7/872)	A2063G (100%)	Kogoj et al., 2018 [27]

**Table 3 jcm-11-01782-t003:** The features of commercial kits to detect MRMP.

Assay	Manufacturer	Target Gene Related with MRMP	Reference
LightMix^®^	Roche	23 S rRNA (2063, 2064)	[30]
GENECUBE^®^ Mycoplasma	TOYOBO	23 S rRNA (2063, 2064, 2067)	[34]
Smart Gene^®^	Mizuho Medy	23 S rRNA (2063, 2064)	[35]

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
