# Peer review of "Recent Trends in the Epidemiology, Diagnosis, and Treatment of Macrolide-Resistant Mycoplasma pneumoniae"

_jcm, 2022, doi:10.3390/jcm11071782_

Round 1

Reviewer 1 Report

*INTRODUCTION
- ref.2 to 7 could be reduced as regards lines 20-22. Moreover, there are some inconsistencies in the related references number. Based on that, I think ref. 5-7 can be removed. Additionally, the reference style of the remaining ones should be corrected according to the journal style. 
- In general, I think the introduction should be expanded. 
- First, the authors should define the MP-related disease spectrum and, inside it, should also mention the fact that MP has been also implicated in extra-pulmonary disorders, especially in children (Curr Opin Rheumatol
. 2018 Jul;30(4):380-387). 
- In this regard, recent articles highlighted that the delay in the MP appropriate antimicrobial treatment, which can be also due to its antibiotic resistance, can be implicated in the severity of clinical course of respiratory disease (J Microbiol Immunol Infect. 2019 Apr;52(2):329-335) and, notably, even extra-pulmonary disorders (J Microbiol Immunol Infect. 2020 Feb;53(1):188-189. doi: 10.1016/j.jmii.2019.04.011)
- some general epidemiological data related to difference between children and adults may be also useful as a background.
- Overall, the introduction should be improved, expanded a little and updated with some important concepts related to the MP diseases spectrum.

*2. Mechanisms of macrolide resistance for M.pneumoniae
- I would suggest the authors to expand a little this section too, by providing more details related to the mechanisms of action for the different classes of antibiotics used against MP.
- Moreover, some specifications about the main macrolides (ERY, CLA and AZI) would be useful for the readers, in my opinion. 

*Antimicrobial susceptibility of macrolide-resistant M.pneumoniae
- Table 1 is linked to a specific reference. Is it the same table? Did the authors get authorization for reproduction? Similar questions for table 2.
- Unfortunately, there is confusion with the references numbers: there is no correspondence apparently and, again, the reference style is not consistent with the journal policy and some references are even incomplete for some reasons. 
- Moreover, I would suggest the authors to make new tables summarizing the most relevant studies rather than their own one only. 

*4. Diagnosis
- “Bacterial culture of MP is necessary for diagnosis.” As it is, this sentence is definitely wrong. The diagnosis of MP infection can rely on serological, antigenic and/or PCR-based tests. Bacterial culture is not pursued at all. Probably, the authors refer to the assessment of MP antibiotic resistance?
- I think this section should focus better on the methods available for assessment of MP antibiotic resistant strain. A table comparing the different methods would be useful

*5. Treatment
- Unlike the previous sections, which are relatively short, this section is too long. Rather than making a general discussion on antibiotic therapy in MP, the authors should focus on the approach and options for those patients resulting MP antibiotic resistant (e.g. Eur J Clin Microbiol Infect Dis. 2019 Apr;38(4):631-635. doi: 10.1007/s10096-018-03448-0; Expert Rev Anti Infect Ther. 2018 Jan;16(1):23-34. doi: 10.1080/14787210.2018.1414599, others)

*5.2. Therapy other than antibiotics: systemic corticosteroids
- I think this section is not pertinent to the main topic of the review.

*Conclusions
- unfortunately, I cannot see clear conclusions and the paper seems really focused on the Japanese setting, which is not consistent with the title. 

*References
- there are several inconsistencies with the numeration, the reference style is not appropriate.
- in general, the references are incomplete and need a deeper literature research. 

Author Response

Dear Reviewer 1,

Thank you very much for your review of our paper and constructive comments. We have revised the paper based on your comments.

Therefore, please check again.

*INTRODUCTION

  • Point 1:ref.2 to 7 could be reduced as regards lines 20-22. Moreover, there are some inconsistencies in the related references number. Based on that, I think ref. 5-7 can be removed. Additionally, the reference style of the remaining ones should be corrected according to the journal style.  
  • Point 1: Thank you for the indication. We have removed ref. 5-7 as you indicated.
  • Point 2:In general, I think the introduction should be expanded. 
  • Point 2: Thank you for your suggestion. We have expanded the introduction section accordingly.
  • First, the authors should define the MP-related disease spectrum and, inside it, should also mention the fact that MP has been also implicated in extra-pulmonary disorders, especially in children (Curr Opin Rheumatol
    . 2018 Jul;30(4):380-387). 
  • Point 3: Thank you for your advice. We have added the sentence in line 20 and the reference.
  • In this regard, recent articles highlighted that the delay in the MP appropriate antimicrobial treatment, which can be also due to its antibiotic resistance, can be implicated in the severity of clinical course of respiratory disease (J Microbiol Immunol Infect. 2019 Apr;52(2):329-335) and, notably, even extra-pulmonary disorders (J Microbiol Immunol Infect. 2020 Feb;53(1):188-189. doi: 10.1016/j.jmii.2019.04.011)
  • Point 4: Thank you for your comment. We have added the sentences in lines 28-29, as well as the references.
  • some general epidemiological data related to difference between children and adults may be also useful as a background.
  • Point 5: Thank you for the indication. We have added the sentence in lines 23-24 and the reference.
  • - Overall, the introduction should be improved, expanded a little and updated with some important concepts related to the MP diseases spectrum
  • Point 6: Thank you for your suggestion. We have improved and expanded the introduction section, as per your advise. 

*2. Mechanisms of macrolide resistance for M.pneumoniae

  •  I would suggest the authors to expand a little this section too, by providing more details related to the mechanisms of action for the different classes of antibiotics used against MP.
  • Point 7: Thank you for your advice. We have expanded this section to provided more information.
  • - Moreover, some specifications about the main macrolides (ERY, CLA and AZI) would be useful for the readers, in my opinion
  • Point 8: Thank you for your comment. We have explained it in lines 41-43, and 45-53, and added the references, as per your suggestion.
  • - Unfortunately, there is confusion with the references numbers: there is no correspondence apparently and, again, the reference style is not consistent with the journal policy and some references are even incomplete for some reasons. 
  • Point 9: Thank you for your comment. The reference number has been corrected.
  • - Moreover, I would suggest the authors to make new tables summarizing the most relevant studies rather than their own one only. 
  • Point 10: Thank you for your suggestion. We have created a new Table (Table. 1) including to the new literature and have removed the old ones.

*4. Diagnosis

  • - “Bacterial culture of MP is necessary for diagnosis.” As it is, this sentence is definitely wrong. The diagnosis of MP infection can rely on serological, antigenic and/or PCR-based tests. Bacterial culture is not pursued at all. Probably, the authors refer to the assessment of MP antibiotic resistance?
  • Point 11: Thank you for your indication. We have revised the sentence in lines 83-85 accordingly.
  • - I think this section should focus better on the methods available for assessment of MP antibiotic resistant strain. A table comparing the different methods would be useful
  • Point 12: Thank you for your comment. We have made a new table (Table. 3) and added information on the method for assessment of MRMP resistance.

*5. Treatment

  • - Unlike the previous sections, which are relatively short, this section is too long. Rather than making a general discussion on antibiotic therapy in MP, the authors should focus on the approach and options for those patients resulting MP antibiotic resistant (e.g. Eur J Clin Microbiol Infect Dis. 2019 Apr;38(4):631-635. doi: 10.1007/s10096-018-03448-0; Expert Rev Anti Infect Ther. 2018 Jan;16(1):23-34. doi: 10.1080/14787210.2018.1414599, others)
  • Point 13: Thank you for your advice. We have added the references and sentences in lines 171-176 as suggested
  • - Unlike the previous sections, which are relatively short, this section is too long. Rather than making a general discussion on antibiotic therapy in MP, the authors should focus on the approach and options for those patients resulting MP antibiotic resistant (e.g. Eur J Clin Microbiol Infect Dis. 2019 Apr;38(4):631-635. doi: 10.1007/s10096-018-03448-0; Expert Rev Anti Infect Ther. 2018 Jan;16(1):23-34. doi: 10.1080/14787210.2018.1414599, others)
  • Point 14: Thank you for your comment. This section is not the main section as you suggested. Therefore, we have removed the sentence and shortened it.

*Conclusions

  • - unfortunately, I cannot see clear conclusions and the paper seems really focused on the Japanese setting, which is not consistent with the title.
  • Point 15: Thank you for your advice. We have removed the description of Japan as suggested

*References

  • - there are several inconsistencies with the numeration, the reference style is not appropriate.
  • - in general, the references are incomplete and need a deeper literature research. 
  • Points 16 and 17: Thank you for your comments. We have corrected these aspects of the manuscript.

Reviewer 2 Report

Dear authors,

The manuscript „Recent trends in the epidemiology, diagnosis, and treatment of macrolide-resistant Mycoplasma pneumoniae” is well written, it properly describes the epidemiology, diagnosis and treatment of M. pneumoniae. The recommendations for use the antibiotics are important for further analysis. The conclusion is important in  the treatment of this pathogen.

References should be corrected, the numbers are not proper from number 6.

Authors should check if they always use italic in “M. pneumoniae” (table 1, 2, line 28 and elsewhere).

Line 23: “these differences remain unclear”: it would be profitable to point some reasons.

Line 28: Add space in “M.pneumoniae”

Table 1: Add the information which are the reference strains.

Diagnosis: It would be profitable to add the information from what material particular types of tests can be made.

Line 92: Change “machines” into “equipment”.

Line 121: Add the number of reference – Lung et al.

Line 141: Add space before “are”.

Author Response

Dear Reviewer 2

Thank you for your review. I revised as you indicated. Therefore, I would like you to check it again.

  • References should be corrected, the numbers are not proper from number 6.
  • Point 1: Thank you for your comment. We have corrected this.
  • Authors should check if they always use italic in “M. pneumoniae” (table 1, 2, line 28 and elsewhere).
  • Point 2: Thank you for your indication. We have ensured consistency in italicization.
  • Line 23: “these differences remain unclear”: it would be profitable to point some reasons.
  • Point 3: Thank you for your suggestion. We have revised this sentence, as seen in lines 25-26.
  • Add space in “M.pneumoniae”
  • Point 4: Thank you for your comment. We have corrected this error.
  • Table 1: Add the information which are the reference strains.
  • Point 5: Thank you for your suggestion. We have revised Table. 1 and added information in the reference section.
  • Diagnosis: It would be profitable to add the information from what material particular types of tests can be made.
  • Point 6: Thank you for your comment. We have added the information on materials in lines 93-94.
  • Line 92: Change “machines” into “equipment”.
  • Line 121: Add the number of reference – Lung et al.
  • Line 141: Add space before “are”.

Point 7-9: Thank you for your comments. We have corrected this error.

Round 2

Reviewer 1 Report

The authors significantly improved the manuscript, but I still have several major points that should be addressed, as explained below. 

- Old table 2 is probably remained in the main manuscript along with the new table. Please, fix it.
-There are several grammar inconsistencies and typing mistakes, even in Table 3 (“reffrence”).
- DIAGNOSIS section: I think this section is still too focused on the microbiological diagnosis of MP infection in general (where some conceptual mistakes have been corrected), but the part about the diagnosis (or better detection, since this is not a common clinical practice, of course) of antimicrobial resistance has not been developed enough yet, which is indeed the core topic of this review. 
- Since the title is “Recent trends in the epidemiology, diagnosis, and treatment of  macrolide-resistant Mycoplasma pneumoniae”, I think the authors should also include a section discussing the (potential) impact of COVID-19 on the spread of antibiotic resistance of MP against macrolides, who have been widely used to treat COVID-19 or other viral infections, in order to prevent pneumonia with bacterial superimposition or because of some potential antiviral (not supported enough: “Clinical evidences on the antiviral properties of macrolide antibiotics in the COVID-19 era and beyond”) or anti-inflammatory properties (“Nonantimicrobial Actions of Macrolides: Overview and Perspectives for Future Development”). I think this discussion would be useful to the readership, quite timely and may increase the reference to this paper in future. 
- In terms of epidemiology, I think the authors could make a discussion about differences in MP macrolide resistance between adults and children, after an appropriate literature search.
- Also, some considerations comparing developing and developed countries may be appropriate, if there are specific research addressing this question.
- References to be updated and completed according to the new discussion points and sections suggested above.

Author Response

Response to Reviewer 1 Comments

Dear Reviewer 1,

Thank you very much for your review of our paper and constructive comments. We have revised the paper based on your comments.

Therefore, please check again.

*INTRODUCTION

Point 1:ref.2 to 7 could be reduced as regards lines 20-22. Moreover, there are some inconsistencies in the related references number. Based on that, I think ref. 5-7 can be removed. Additionally, the reference style of the remaining ones should be corrected according to the journal style.  

Point 1: Thank you for the indication. We have removed ref. 5-7 as you indicated.

Point 2:In general, I think the introduction should be expanded. 

Point 2: Thank you for your suggestion. We have expanded the introduction section accordingly.

First, the authors should define the MP-related disease spectrum and, inside it, should also mention the fact that MP has been also implicated in extra-pulmonary disorders, especially in children (Curr Opin Rheumatol
. 2018 Jul;30(4):380-387). 

Point 3: Thank you for your advice. We have added the sentence in line 20 and the reference.

In this regard, recent articles highlighted that the delay in the MP appropriate antimicrobial treatment, which can be also due to its antibiotic resistance, can be implicated in the severity of clinical course of respiratory disease (J Microbiol Immunol Infect. 2019 Apr;52(2):329-335) and, notably, even extra-pulmonary disorders (J Microbiol Immunol Infect. 2020 Feb;53(1):188-189. doi: 10.1016/j.jmii.2019.04.011)

Point 4: Thank you for your comment. We have added the sentences in lines 28-29, as well as the references.

some general epidemiological data related to difference between children and adults may be also useful as a background.

Point 5: Thank you for the indication. We have added the sentence in lines 23-24 and the reference.

- Overall, the introduction should be improved, expanded a little and updated with some important concepts related to the MP diseases spectrum

Point 6: Thank you for your suggestion. We have improved and expanded the introduction section, as per your advice. 

*2. Mechanisms of macrolide resistance for M.pneumoniae

I would suggest the authors to expand a little this section too, by providing more details related to the mechanisms of action for the different classes of antibiotics used against MP.

Point 7: Thank you for your advice. We have expanded this section to provided more information.

- Moreover, some specifications about the main macrolides (ERY, CLA and AZI) would be useful for the readers, in my opinion

Point 8: Thank you for your comment. We have explained it in lines 41-43, and 45-53, and added the references, as per your suggestion.

- Unfortunately, there is confusion with the references numbers: there is no correspondence apparently and, again, the reference style is not consistent with the journal policy and some references are even incomplete for some reasons. 

Point 9: Thank you for your comment. The reference number has been corrected.

- Moreover, I would suggest the authors to make new tables summarizing the most relevant studies rather than their own one only. 

Point 10: Thank you for your suggestion. We have created a new Table (Table. 1) including to the new literature and have removed the old ones.

*4. Diagnosis

- “Bacterial culture of MP is necessary for diagnosis.” As it is, this sentence is definitely wrong. The diagnosis of MP infection can rely on serological, antigenic and/or PCR-based tests. Bacterial culture is not pursued at all. Probably, the authors refer to the assessment of MP antibiotic resistance?

Point 11: Thank you for your indication. We have revised the sentence in lines 83-85 accordingly.

- I think this section should focus better on the methods available for assessment of MP antibiotic resistant strain. A table comparing the different methods would be useful

Point 12: Thank you for your comment. We have made a new table (Table. 3) and added information on the method for assessment of MRMP resistance.

*5. Treatment

- Unlike the previous sections, which are relatively short, this section is too long. Rather than making a general discussion on antibiotic therapy in MP, the authors should focus on the approach and options for those patients resulting MP antibiotic resistant (e.g. Eur J Clin Microbiol Infect Dis. 2019 Apr;38(4):631-635. doi: 10.1007/s10096-018-03448-0; Expert Rev Anti Infect Ther. 2018 Jan;16(1):23-34. doi: 10.1080/14787210.2018.1414599, others)

Point 13: Thank you for your advice. We have added the references and sentences in lines 168-173 as suggested

- Unlike the previous sections, which are relatively short, this section is too long. Rather than making a general discussion on antibiotic therapy in MP, the authors should focus on the approach and options for those patients resulting MP antibiotic resistant (e.g. Eur J Clin Microbiol Infect Dis. 2019 Apr;38(4):631-635. doi: 10.1007/s10096-018-03448-0; Expert Rev Anti Infect Ther. 2018 Jan;16(1):23-34. doi: 10.1080/14787210.2018.1414599, others)

Point 14: Thank you for your comment. This section is not the main section as you suggested. Therefore, we have removed the sentence and shortened it.

*Conclusions

- unfortunately, I cannot see clear conclusions and the paper seems really focused on the Japanese setting, which is not consistent with the title.

Point 15: Thank you for your advice. We have removed the description of Japan as suggested

*References

- there are several inconsistencies with the numeration, the reference style is not appropriate.

- in general, the references are incomplete and need a deeper literature research. 

Points 16 and 17: Thank you for your comments. We have corrected these aspects of the manuscript.
